# Digital Image Speckle Correlation (DISC): Facial Muscle Tracking for Neurological and Psychiatric Disorders

**DOI:** 10.3390/diagnostics15131574

**Published:** 2025-06-20

**Authors:** Shi Fu, Pawel Polak, Susan Fiore, Justin N. Passman, Raphael Davis, Lucian M. Manu, Miriam Rafailovich

**Affiliations:** 1Department of Materials Science and Chemical Engineering, Stony Brook University, Stony Brook, NY 11794, USA; shi.fu@stonybrook.edu; 2Department of Applied Mathematics and Statistics, Stony Brook University, Stony Brook, NY 11794, USA; pawel.polak@stonybrook.edu; 3Department of Neurosurgery, Renaissance School of Medicine, Stony Brook University, Stony Brook, NY 11794, USA; susan.fiore@stonybrookmedicine.edu (S.F.); justin.passman@stonybrookmedicine.edu (J.N.P.); raphael.davis@stonybrookmedicine.edu (R.D.); 4Department of Psychiatry, Renaissance School of Medicine, Stony Brook University, Stony Brook, NY 11794, USA; lucian.manu@stonybrookmedicine.edu

**Keywords:** digital image speckle correlation, facial muscle tracking, non-contact monitoring, reaction time assessment

## Abstract

**Background/Objectives**: Quantitative assessments of facial muscle function and cognitive responses can enhance the clinic evaluations in neuromuscular disorders such as Bell’s palsy and psychiatric conditions including anxiety and depression. This study explored the application of Digital Image Speckle Correlation (DISC) in detecting enervation of facial musculature and assessing reaction times in response to visual stimuli. **Methods**: A consistent video recording setup was used to capture facial movements of human subjects in response to visual stimuli from a calibrated database. The DISC method utilizes the displacement of naturally occurring skin pores to map the specific locus of underlying muscular movement. The technique was applied to two distinct case studies: Patient 1 had unilateral Bell’s palsy and was monitored for 1 month of recovery. Patient 2 had a comorbidity of refractory depression and anxiety disorders with ketamine treatment and was assessed over 3 consecutive weekly visits. For patient 1, facial asymmetry was calculated by comparing left-to-right displacement signals. For patient 2, visual reaction time was measured, and facial motion intensity and response rate were compared with self-reported depression and anxiety scales. **Results**: DISC effectively mapped biomechanical properties of facial motions, providing detailed spatial and temporal resolution of muscle activity. In a control cohort of 10 subjects, when executing a facial expression, the degree of left/right facial asymmetry was determined to be 13.2 (8)%. And showed a robust response in an average of 275 (81) milliseconds to five out of the five images shown. For patient 1, obtained an initial asymmetry of nearly 100%, which decreased steadily to 20% in one month, demonstrating a progressive recovery. Patient 2 exhibited a prolonged reaction time of 518 (93) milliseconds and reduced response rates compared with controls of 275 (81) milliseconds and a decrease in the overall rate of response relative to the control group. The data obtained before treatment in three visits correlated strongly with selected depression and anxiety scores. **Conclusions**: These findings highlight the utility of DISC in enhancing clinical monitoring, complementing traditional examinations and self-reported measures.

## 1. Introduction

Nerve stimulation of facial muscles (neuromuscular innervation) is essential for controlling facial movements and expressions. Subtle alterations in facial muscle activation patterns can serve as indicators of underlying pathologies, including peripheral nerve damage [1], central nervous system issues, or certain psychiatric conditions [2]. Popular ways of monitoring facial muscle enervation often involve direct measurement of muscular activity such as surface electromyography (sEMG) [3,4,5], which requires physical sensors placement on the face or head.

Digital image speckle correlation (DISC) is a technique originally developed for mapping the deformations of materials [6], where the speckles or random scatterers are distributed across the materials’ surface to allow mapping of surface deformations under a defined force field. It was previously shown that this technique could also be adapted to biomechanical measurements of living tissue [7], but the addition of scattering particles made the method difficult to implement. Staloff et al. showed that the pores on skin provided a natural array of random scatterers, which could be used to profile skin mechanics and demonstrate the deteriorating effects of skin mechanics induced by natural ageing [8]. In humans, the skin on the face is known to be directly attached to the underlying musculature via the Superficial Musculoaponeurotic System (SMAS) [9], and, hence, in addition to superficial elasticity, the technique was shown to be a direct indication of muscular enervation. The technique was first applied as a method for facial recognition, where we demonstrated that it was another method for determination of the muscles involved in facial expression, which are unique to the individual [10]. A much larger 18 months trial was performed on patients injected with Botox, where Bhatnagar et al. demonstrated that DISC can follow the location and intensity of the paralysis as well as the subsequent recovery [11]. Verma et al. demonstrated that DISC can be used to measure the intensity of motion required for a given expression, which is unique to the individual [12]. Using this information, DISC was used to determine the optimal dose of Botox required to achieve muscular paralysis. This study also showed improved recovery in patients where DISC was used to guide the injection site, relative to another group where injection was performed following standard protocols.

Most recently, Saadon et al. showed that emotional response could also be detected via rapid facial muscular enervation, without any external facial expression, and in this manner it could be used for real-time detection and classification of micro-emotions (happy, sad, and neutral) by analyzing subtle muscular features invisible to the naked eye [13].

The application of DISC to the analysis of a specific pathology was demonstrated by Bhatnagar, where it was shown that DISC could detect impairment of the facial nerve and subsequent recovery following surgery, in patients diagnosed with vestibular schwannoma [14].

Except for the acoustic neuroma study, the previous work selected only healthy individuals, where the experiments were conducted mostly to demonstrate the abilities of disc in neuromuscular evaluation. Hence, in this paper we selected two pathologies, one involving a direct physical impairment, Bell’s Palsy, which was selected since paralysis and recovery of the affected site could be compared on the same patient. A second patient was selected with a condition involving a mental state, where the response of a patient undergoing specific therapy could be directly compared to a much larger control cohort of healthy individuals. We detail the findings of the technique as compared to the results obtained from a control cohort without underlying pathology to illustrate the power of this non-contact method for diagnosis, accomplished solely through video recording and programming-based analysis.

## 2. Materials and Methods

### 2.1. Human Study Design

The study design comprised three main parts: (1) a control cohort of healthy individuals as the baseline; (2) a case study of a patient with Bell’s palsy, to quantify facial asymmetry in the healing progress; (3) A case study of a patient who is undergoing active ketamine therapy for treatment-refractory depression and anxiety, to assess facial muscle response and reaction time.

All participants and patients provided informed consent, acknowledging both the idea of the DISC technique and the study’s objectives.

#### 2.1.1. Control Groups

The control group consisted of 10 volunteers, ranging from 23 to 56 years of age. The study was conducted under the supervision of the Stony Brook University Committee on Research in Human Subjects (IRB2019-0199). None of the participants in the control group had any known neurological or facial muscle impairments, nor any psychiatric conditions requiring clinical intervention. Exclusion criteria included recent facial surgery, significant dermatological issues affecting the face, or any known nerve injury.

#### 2.1.2. Patient 1 (Bell’s Palsy)

This is a 44-year-old woman with sudden onset of isolated, left-sided Bell’s (cranial nerve VII) Palsy in the setting of resolving Herpes Zoster infection, treated with topical acyclovir. Medical history is otherwise non-contributory. The study timeline commenced at one week after the initial onset of paralysis. Four weekly visits were scheduled, allowing the capture of sequential video recordings to document the evolution of facial asymmetry as she recovered.

#### 2.1.3. Patient 2 (Ketamine Therapy)

This is a 30-year-old woman with a past psychiatric history of treatment-resistant major depressive disorder (TRD), generalized anxiety disorder, and social anxiety disorder receiving ketamine therapy for treatment-refractory depression. Medical history is otherwise non-contributory. The patient was initiated on intranasal Esketamine at 28 mg 1×/2 weeks with Meclizine 50 mg PO induction. This is in addition to her baseline psychopharmacological regimen of Zoloft, Lybalvi, Seroquel, Gabapentin, and as needed Xanax. During this period, the patient has reported drastic improvement in her depressive, mood, and anxiety symptoms with no serious adverse effects. This regimen has continued for the six years preceding our recordings. This study consisted of observations during three consecutive weeks’ visits, where we obtained reaction data, as described below, capturing recordings both before and immediately after ketamine treatment sessions.

### 2.2. Video Recording Setup

The standardized setup ensured consistent data acquisition across control group and patient case studies (Figure 1a schematic). The subject’s head was placed on a cushioned stationary stand, a computer screen placed from 20 inches away at eye level, and a mirror was placed below the subject’s head where the stimulus slides shown on the screen were reflected. This setup synchronizes the onset of the visual stimulus and any detectable facial reaction. The video was recorded at 4K resolution with 60 fps, using an iPhone15 Pro or a Canon Rebel, in the manual focusing mode ensuring that skin pores were consistently in clear view.

### 2.3. Stimulus

The stimulus images were selected from the International Affective Picture System (IAPS) library [15], focusing specifically on valence and arousal ratings for female subjects. Each stimulus image was chosen for its similar arousal level (5.1–6.1). Images associated with high valence (valence > 7.8) were used as stimuli; the detailed selection is shown Table 1. To establish a neutral baseline before the test, a blank image was shown immediately prior to the first stimulus. The presentation consisted of five stimulus images, each displayed for 10 s before transitioning to the next.

For each stimulus, the subjects’ recognition could be detected evidently by subtle facial muscle changes. Meanwhile, any lack of response was logged as a “dissociation” with the stimulus.

### 2.4. DISC Algorithm

The DISC algorithm has been extensively described in numerous previous publications [16,17,18]. Briefly, DISC compares two frames via the optical flow patterns, which describe the motion caused by the movement of speckles, in our case, skin pores. It operates under two basic assumptions: (1) the pixel intensities of a small area remain constant between frames, and (2) neighbouring pixels have similar motion. The Lucas–Kanade method, widely used for estimating optical flow requires constant flow in a local pixel neighbourhood and solves for flow parameters to best align this neighbourhood between frames [19]. This differential approach uses image gradient information to generate the vectors describing the local motion data. The motion field can then be presented as a heatmap corresponding to the speckle’s displacement or the vector amplitude.

### 2.5. Active Muscle Regions

Since the individual motions are sometimes too small and occur too fast for visual detection, the motion field from each frame undergoes a square root transformation. By applying the square root to each data point, larger values are scaled down more significantly than smaller ones, reducing the range and variance in visual analysis. The active facial area responding to stimuli is then determined by averaging the transformed values at each location throughout the stimulus period.

Figure 1b represents a consolidated view of the active muscle regions. It highlights the regions that shift and deform in response to specific localized muscle movements triggered by the stimulus. A composite region can then be formed via the addition of all the individual time sequence images showing the area of enervation involved in this response. The motion intensity can then be determined by averaging the speckle’s displacement within the active muscle regions at each time frame.

### 2.6. Reaction Time Measurements

The reaction time is defined as the interval from the beginning of a new stimulus to the initiation of facial muscle motion. By taking the derivative of the motion intensity over time, revealing the first peak (acceleration of motion) represents the timepoint of cognitive recognition, hence reaction time. In this study, the time resolution is 1/60s, which is limited by the frame rate of the camera.

### 2.7. Statistical Analysis

All analysis in this study were conducted by implementation of Python 3.12.0 and its libraries. OpenCV and Pillow facilitate image frames extraction and motion tracking, which were required for DISC analysis. SciPy, Pandas, and Scikit-Learn were used for statistics, including *t*-tests, ANOVA, and correlation analyses. Matplotlib 3.10 was used for result visualization. Statistical significance was set at *p* < 0.05.

## 3. Results

### 3.1. Control Group

For the control group, as depicted in Figure 2a, the plot of left vs. right face motion aligning the intercept with the origin. The linearity with a slope near 1 signifies symmetry between the left and right sides of the face indicates that participants in the control group exhibit minimal deviation in muscle activity across the two sides of the face during expressions.

Figure 2b illustrates facial motion intensity development with the five stimuli in sequence. The initial 5 s (negative time) correspond to a blank image, during which no response is observed. At time zero, when the first image appears, a sharp rise is detected, indicating a muscular response or arousal. The analysis reveals that at the shown frequency, the facial motion decay does not consistently return to baseline, allowing arousal from one stimulus to overlap with the next. The visual reaction time (VRT) was then calculated averaging the reaction time the five stimuli shown. The average VRT obtained for the control cohort was approximately t_r_ 275 (81) milliseconds.

### 3.2. Case Study 1: Patient Diagnosed with Bell’s Palsy

A simple case illustrating the ability of DISC to track muscle movement occurs in the analysis of the facial paralysis induced in patient 1 diagnosed with Bell’s Palsy. Since paralysis is localized to one side of the face (left), comparison with the unaffected side provides a direct method for probing the efficacy of the DISC method.

In Figure 3a, we show the images obtained weekly of the patient with the DISC heatmaps overlayed. For the first week, the left side of the face shows no muscular movement at all when the patient attempts to produce a small smiling motion. By the second week, faint stirring is evident, while after the third week significant mobility has resumed. Finally, in the fourth week, relative symmetry in the extent of motion has returned.

The degree of asymmetry (DA) can be quantified via integrating the motion data on each side of the face and taking the difference, then normalizing by the stronger side to account for individual differences (Equation (1)):DA = |Left − Right|/max (Right, Left) × 100%,(1)

For the control group, we calculated DA = 13.2 (8)%, which is averaged over the ten subjects, and we plotted it in Figure 3b as a normative value compared to patient 1. For patient 1, we plotted DA as a function of the time after diagnosis where we find that the value continuously decreases, approaching the value of the control group after four weeks, indicating the recovery of facial nerve function.

### 3.3. Case Study 2: Patient Undergoing Ketamine Treatment

At each visit, prior to ketamine administration, patient 2 was evaluated for depression and anxiety using the scales described in Table 2, allowing for a direct comparison between physiological responses captured by DISC and subjective reports of depression/anxiety symptoms. Self-reported scores indicated persistent but moderate symptoms of depression and moderate-to-severe anxiety.

The same stimulus images presented to the control group were also shown to patient 2. Figure 4a illustrates patient 2’s facial motion intensity over time for each of the five images, comparing responses before and after ketamine treatment. The derivative of the motion intensity determines the onset of reaction to the stimulus, hence the reaction time. For some stimulus, no discernable muscle activation was observed to a given stimulus, indicating a dissociative instance. Table 2 summarizes the number of dissociative responses recorded for each visit before and after treatment.

In Figure 4b, we show the facial heat maps corresponding to the sum of the reactions to all five images shown in each visit prior and after the administration of ketamine. It is interesting to note that the initial intensity of the reaction decreased in each consecutive visit prior to treatment. After treatment, except for the first visit, where a large decrease is observed, a small increase is observed in second and third visits. Prior to treatment, in the first two visits, the most active muscular movement in response to the visual stimulus is concentrated on the left cheek, while following ketamine administration, the movement patterns become more evenly distributed across the face, and the previously intense activity in the left cheek is dissipated. This shift suggests a relaxation effect induced by ketamine, reducing localized muscle tension and promoting a more balanced facial muscle response. On the third visit, the initial muscular motion is low, and only a small increase, distributed across the face, is observed following treatment.

## 4. Discussion

DISC offers a fully non-contact method in measuring neuromuscular enervation, enabling remote self-monitoring by the patient with minimal discomfort. The data for analysis is obtained from video footage using commercially available camera or smartphones, and, hence, it provides a cost-effective and user-friendly solution, which eliminates the need for specialized hardware. Furthermore, the setup is portable and can be implemented at bedside or in the field, where other specialized medical equipment may not be available.

It is worth noting that this paper describes two case studies, where the applicability of the technique is described in each case, based on the physical principles of the methodology. This is not a population study, and, hence, we cannot predict general outcomes.

### 4.1. Facial Paralysis Evaluation

Previous studies have demonstrated that DISC can map the intensity of muscular motion across different regions of the face [10,13]. For example, using a Botox injection model, Bhatnagar et al. monitored the location of the paralysis following injection and the recruitment of alternative muscles in enabling facial expression in otherwise healthy patients [11].

Here, we extended our application of the technique to detect paralysis associated with a specific pathology. We show that the absolute intensity of the motion required to exhibit a comparable expression will vary greatly between individuals in the control cohort, yet, the degree of asymmetry, between the right and left sides of the face, normalized by the intensity, is less than 13.5%.

In case study 1, we tracked the recovery progress of a patient diagnosed with Bell’s Palsy and quantified the degree of pathology relative to the response of the normal cohort. In Figure 3b, we found that after only four weeks it reached the baseline of the normal control group (normative value), indicating good recovery in all segments of the facial area. Traditional clinical scales for facial paralysis like the House–Brackmann grading system [20] remain somewhat subjective and lack the spatial detail or real-time quantitative evaluation that DISC can provide. Moreover, the ability to adequately discern paralysis in different areas also theoretically allows DISC to provide rapid and objective diagnostic insights such as differentiating stroke from Bell’s palsy by analyzing the location of facial muscle innervation. Bell’s palsy is a temporary condition causing facial paralysis on one side of the face due to inflammation of cranial nerve VII, which can weaken or paralyze both upper and lower facial muscles, while stroke typically involves lower motor neurons, impacting the entire side of the face but sparing the forehead because of central nervous system damage. This capability allows families and medical professionals to take immediate and appropriate action, ensuring strokes receive the urgent medical attention they require, while Bell’s palsy, though significant, can be managed without emergency intervention.

### 4.2. Psychiatric Evaluations

Traditional psychiatric evaluations rely heavily on clinical interviews and questionnaires, which depend on the patient’s self-report and the clinician’s interpretation. However, these methods are often hindered by low motivation, lack of insight, or memory biases, leading to incomprehensive assessments. By identifying subtle discrepancies between reported and observed reactions, DISC can enhance the understanding of a patient’s mental and emotional state, offering insights into delayed responses or heightened reactivity.

#### 4.2.1. Reaction Time

Reaction time has long been a basic measure in psychology and neurology. Various methods, such as catching a projectile [21] or pressing a button [22,23], while accurate, are inherently dependent on the subjects’ physical dexterity. Consequently, individuals with physical impairments may face limitations in participating in such assessments, potentially complicating the analysis. On the other hand, the DISC measures involuntary muscular reactions via video recording, which makes it generally applicable to all patients and even in cases where remote analysis is needed.

From Figure 4c, we find that the reaction time for patient 2 remained relatively unchanged before and after treatment but was significantly slower compared to the control group (*p* < 0.005). The average VRT obtained for control group was approximately 275 with a standard deviation of 81milliseconds, which is in excellent agreement with the values obtained by Jain et al. [23]. These results are consistent with the established range of human reaction times, typically spanning from 200 to 400 milliseconds, depending on factors such as attention, cognitive load, and individual neurological processing speed [22]. Our findings suggest that continual long-term exposure to ketamine may contribute to slower reaction time, potentially due to its effects on neuromuscular control and cognitive processing speed [24]. The influence of long term ketamine treatment on the reaction time of the patients has also been discussed in some studies [25,26]. While delayed reaction times were shown to be correlated with increased dissociation [27] and anxiety in Ketamine treated subjects [28], studies have also been reported on impaired driving and operation of mechanical equipment [29]. Further studies are needed to clarify the extent and mechanisms of ketamine’s impact on engagement and reaction time in patients undergoing prolonged treatment.

#### 4.2.2. Self-Reported Scores vs. DISC Results

Research suggests that quantitative measurements of reaction times and facial movements may be associated with certain aspects of dissociation and cognitive functioning [30,31]. The intensity of facial expressions directly correlates with the level of patient engagement when viewing visual stimuli [32,33,34,35]. These relationships are complex and can vary depending on the specific cognitive domains and clinical conditions being studied. We, therefore, illustrate DISC’s application to a patient who initially presented with treatment resistant MDD and GAD. Table 2 summarizes the number of stimulus that the patient 2 showed no muscle activation, i.e., dissociation. Among control cohort, in contrast, displayed a 100% response rate to the same stimulus. By calculating the Pearson correlation coefficient (r) between the dissociation counts with the patient’s self-report scores from the PHQ-9, QIDS-SR, and BDI assessments, as shown in Figure 4d, strong positive correlations emerged for dissociation with PHQ-9 and QIDS-SR scores, while a moderate positive correlation with BDI. We plot the integrated intensity of her facial reactions to the stimulus, alongside her self-reported scores from the PCL-5, GAD, and BAI assessments. A clear negative correlation with the self-reported PCL-5 and BAI test scores, which show increasing levels of anxiety with each visit. Our findings are in line with other studies that have reported decreased facial expression associated with increased anxiety [36,37], where the extent of the facial expression is in turn correlated with degree of muscular enervation.

Although the patient reported chronic relief from anxiety and depression with her long-term ketamine administration, mild fluctuations remained, evidenced in both her self-report scales and the DISC data. Our observations correspond to a small segment of patient 2’s overall treatment for depression, where her medical records indicate that her condition is fairly stable. Even though we do not have the self-reported scores for these tests after administration of ketamine, the DISC scores are only slightly different, indicating that no drastic changes occurred within the first hour after treatment. The changes we observed in these three visits are small fluctuations in her overall treatment, and, hence, the degree to which they correlate with DISC illustrates the sensitivity of DISC in the detection of the patient’s mental state.

## 5. Conclusions

In conclusion, the application of DISC to the analysis of two distinct case studies of a neurological and psychiatric pathologies demonstrates its clinical potential. Its ability to objectively map facial neuromuscular activity and quantitatively measure reaction time expands its utility across diverse clinical scenarios, which include cognitive and psychological assessments and neurological pathologies affecting the facial nerves. DISC’s non-contact method of operation, ease of use, and cost-effectiveness make it a practical alternative to traditional instrumentation intensive methodology. Hence, DISC can easily be integrated into telemedicine portals and provide rapid diagnosis even under emergency situations.

## Figures and Tables

**Figure 1 diagnostics-15-01574-f001:**
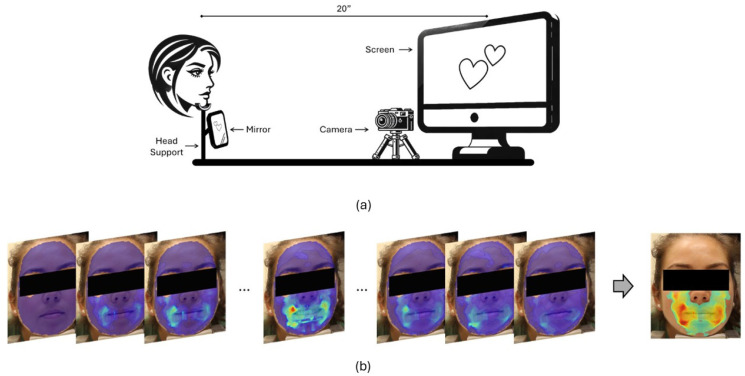
Schematic of data acquisition and analysis for DISC method: (**a**) Video data recording setup showing the head support monitor and mirror used to synchronize the recorded response to appearance of each image. (**b**) Facial motion heatmaps obtained from a sequential set of images following exposure to the stimulus (blue-low -> red-high).

**Figure 2 diagnostics-15-01574-f002:**
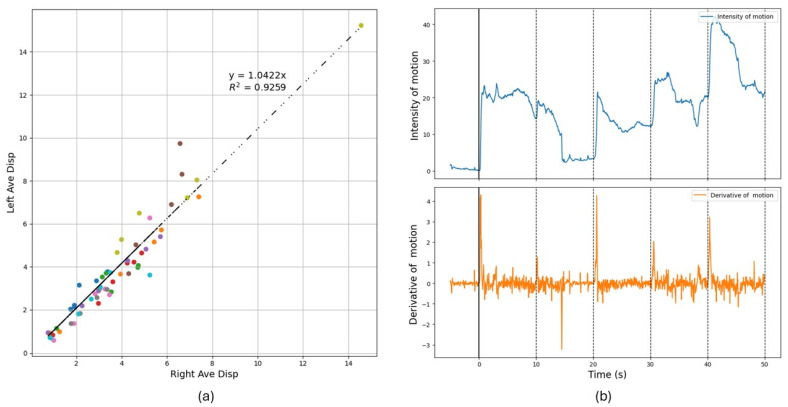
DISC result for control cohort as the baseline. (**a**) Average motion on the left vs. right sides of the face for different moves, for the cohort of control subjects, where each subject appears as a different color. (**b**) Intensity of motion as a function of time in response to the five images shown. Derivative of motion curves indicating the time of reaction acknowledging the stimulus.

**Figure 3 diagnostics-15-01574-f003:**
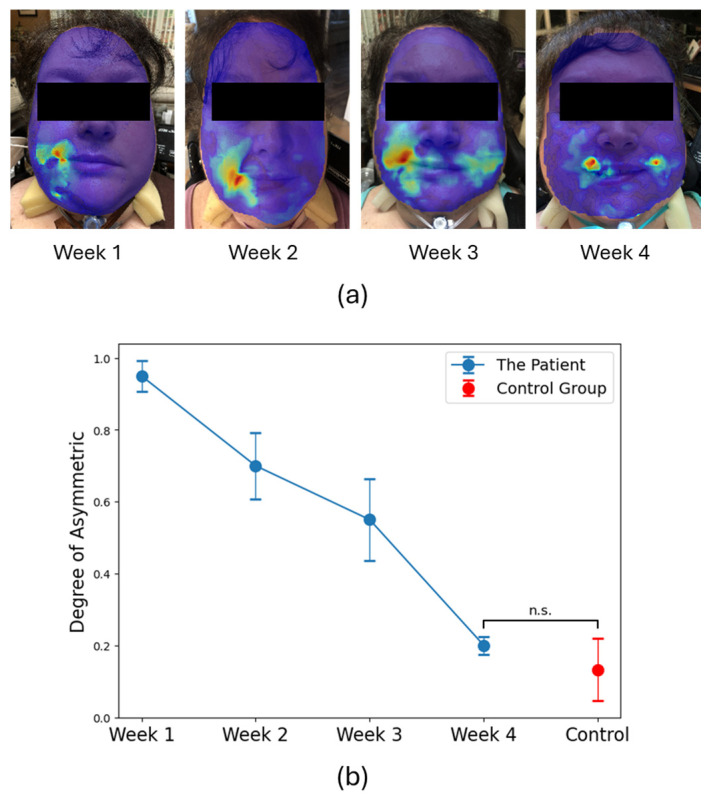
Quantitative assessment for Bell’s Palsy recovery. (**a**) Heatmaps obtained from patient 1 as a function of time from onset of Bell’s Palsy (blue-low to red-high). (**b**) Degree of asymmetry, DA, for each week following diagnosis of patient 1 (blue) together with the DA value for the control cohort (red), as a measure of the recovery process (n.s. indicates no significance).

**Figure 4 diagnostics-15-01574-f004:**
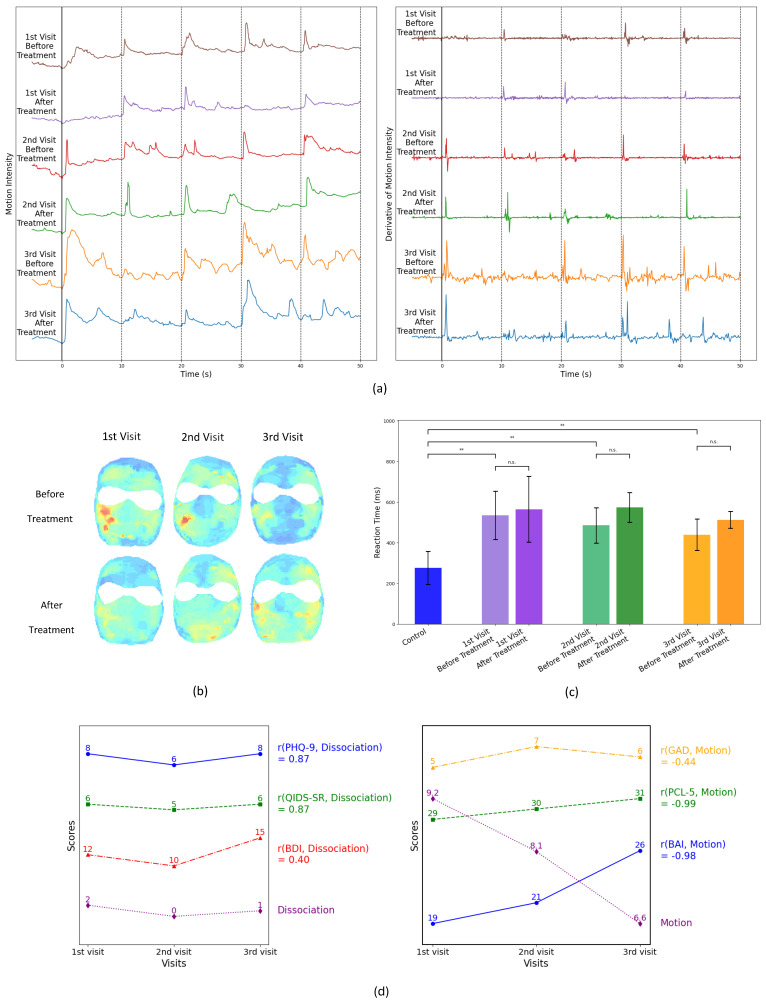
Analysis in response to visual stimuli for patient 2 for each of three visits. (**a**) Intensity of motion as a function of time (blue-low to red-high) and its derivative of motion, indicative of visual reaction time. (**b**) Heatmaps obtained from averaging muscular enervation. (**c**) The reaction time obtained before and after ketamine treatment, together with the average value for the control group (n.s. not significant, ** *p* < 0.01). (**d**) Self-reported scores following each visit with the associated correlation coefficient for depression vs. the level of dissociation and anxiety vs. the motion intensity.

**Table 1 diagnostics-15-01574-t001:** Summary of images shown to the patient of calibrated arousal and valence [15].

	Sequence	Topic	IAPS ID	Valence (std)	Arousal (std)
Before Treatment	1	Kittens	1463	7.81 (1.96)	5.11 (2.17)
	2	Children	2347	8.35 (0.98)	5.88 (2.53)
	3	Bride	2209	7.95 (1.46)	5.91 (2.40)
	4	Fireworks	5910	8.16 (1.15)	5.80 (2.75)
	5	Sailing	8080	7.73 (1.43)	6.25 (2.34)
After Treatment	1	Puppies	1710	8.59 (0.99)	5.31 (2.54)
	2	Baby	2045	8.17 (1.21)	6.02 (2.29)
	3	Wedding	4626	7.80 (1.76)	6.06 (2.51)
	4	Castle	7502	8.15 (1.25)	6.07 (2.58)
	5	Gymnast	8470	7.94 (1.31)	5.98 (2.20)

**Table 2 diagnostics-15-01574-t002:** Data obtained from patient 2 with each visit.

		1st Visit	2nd Visit	3rd Visit
Self-reporting scores prior to ketamine administration				
	PHQ-9: Patient Health Questionaire	8	6	8
	QIDS-SR: Quick Inventory of Depressive Symptomatology-Self-Report	6	5	6
	BDI: Beck Depression Inventory	12	10	15
	BAI: Beck Anxiety Inventory	19	21	26
	GAD: Generalized Anxiety Disorder (GAD-7)	5	7	6
	PCL-5: Posttraumatic Stress Disorder	29	30	31
Number of images for which the patient dissociated				
	Before Treatment	2	0	1
	After Treatment	2	1	2
Visual reaction time(milliseconds): Mean (std)				
	Before Treatment	534 (118)	485 (86)	439 (77)
	After Treatment	564 (161)	573 (74)	512 (42)
Average Intensity of muscular motion in response to the stimulus				
	Before Treatment	9.20	8.05	6.60
	After Treatment	6.74	9.02	7.34

## Data Availability

De-identified excerpts sufficient to replicate the analyses can be provided by the corresponding author upon reasonable request and with approval from the Stony Brook University Committee on Research in Human Subjects.

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
