# Peer review of "Digital Image Speckle Correlation (DISC): Facial Muscle Tracking for Neurological and Psychiatric Disorders"

_diagnostics, 2025, doi:10.3390/diagnostics15131574_

Round 1

Reviewer 1 Report

Comments and Suggestions for Authors

The principal bias of your paper is the very poor sample of patients: only 2!! Why did you decide to write a paper based on only two patients, since these are affected by rare diseases?

Why did you choose a patient affected by refractory depression?

Line 46: what is the meaning?

Line 95: why did you prescribe only topical acyclovir?

Line 106: ...soon after....: you can revise it.

Line 110: why did you list the trade names of the drugs?

Line 269: do you have the normative values for Bell's palsy?

Figure 4: it is poorly readable.

Line 357: DISC's non-contact nature: what is the meaning?

Comments on the Quality of English Language

It is mandatory to improve the English of this paper. 

Author Response

Comments 1: The principal bias of your paper is the very poor sample of patients: only 2!! Why did you decide to write a paper based on only two patients, since these are affected by rare diseases?

Response 1: We thank the reviewer for this important comment. Our intention with this paper was not to present a population-level clinical trial, but to demonstrate the application of DISC to real-world pathologies—building upon years of prior work validating DISC in healthy volunteers and Botox-treated subjects. While the current submission includes only two patients, this was a deliberate choice to highlight two distinct clinical scenarios that could serve as strong proof-of-concept cases: (1) a patient with Bell’s palsy, where facial nerve impairment and recovery could be directly visualized over time and compared within-subject, and (2) a patient with refractory depression undergoing ketamine therapy, which allowed us to correlate subtle neuromuscular signatures with self-reported mental states.

Both patients were selected specifically for their ability to serve as self-controlled studies. In the Bell’s palsy case, facial asymmetry on the paralyzed vs. healthy side provided a clear internal control. In the psychiatric case, changes in facial reaction time and motion intensity could be compared longitudinally within the same patient across treatment sessions and correlated with clinical assessment scores. These designs allowed us to demonstrate the power and sensitivity of DISC in two very different domains—neurology and psychiatry—despite the limited sample size.

We acknowledge that this rational was not stated clearly in the manuscript, and we therefore modified the introduction with a more comprehensive description as follows;

“The technique was first applied as a method for facial recognition, where we demonstrated that it was another method for determination of the muscles involved in facial expression, which are unique to the individual[10]. A much larger 18 months trial was performed on patients injected with Botox, where Bhatnagar et al demonstrated that DISC can follow the location and intensity of the paralysis as well as the subsequent recovery[11]. Verma et al demonstrated that DISC can be used to measure the intensity of motion required for a given expression, which is unique to the individual [12]. Using this information, DISC was used to determine the optimal dose of Botox required to achieve muscular paralysis. This study also showed improved recovery in patients where DISC was used to guide the injection site, relative to another group where injection was performed following standard protocols.

Most recently, Saadon et al showed that emotional response could also be detected via rapid facial muscular enervation, without any external facial expression and in this manner could be used for real-time detection and classification of micro- emotions (happy, sad, and neutral) by analyzing subtle muscular features invisible to the naked eye[13].

The application of DISC to the analysis of a specific pathology was demonstrated by Bhatnagar, where it was shown that DISC could detect impairment of the facial nerve and subsequent recovery following surgery, in patients diagnosed with vestibular schwannoma [14].

 Except for the acoustic neuroma study, the previous work selected only healthy individuals, where the experiments were conducted mostly to demonstrate the abilities of disc in neuromuscular evaluation. Hence in this paper we selected two pathologies, one involving a direct physical impairment, Bell’s Palsy, which was selected since paralysis and recovery of the affected site could be compared on the same patient.  A second patient was selected with a condition involving a mental state, where the response of a patient undergoing specific therapy could be directly compared to a much larger control cohort of healthy individuals.  We detail the findings of the technique as compared to the results obtained from a control cohort without underlying pathology to illustrate the power of this non-contact method for diagnosis, accomplished solely through video recording and programming-based analysis.”

Comment 2: Why did you choose a patient affected by refractory depression?

Response 2: This case thus provided a rare window into the dynamics of emotional processing and dissociation under pharmacological treatment, showcasing the potential of DISC to complement subjective assessments with real-time physiological data. The significance of this case lies in its demonstration of DISC as a tool for psychiatric evaluation, where traditional methods are lacking. The rationale is integrated into the updated introduction section, please see above.

Comment 3: Line 46: what is the meaning?

Response 3: Improved for clarity: “Nerve stimulation of facial muscles (neuromuscular innervation) is essential for controlling facial movements and expressions”

Comment 4: Line 106: ...soon after....: you can revise it.

Response 4: This was revised as follows: removed this statement since prior history is not relevant to the treatment protocol which has been in place for the last six years.

Comment 5: Line 110: why did you list the trade names of the drugs?

Response 5:   The regimen with which she is being treated is relevant to the symptoms we are measuring.  Since the effect of generic medications may vary from those of different trade names, we supplied all the details

Comment 6: Line 269: do you have the normative values for Bell's palsy?

Response 6: In DISC the normative value is the degree of asymmetry in a facial expression, as determined from the average values obtained in the control cohort. In order to make this point clearer we inserted the following in the results section:

Line 226: “For the control group we calculated DA= 13.2 (8)%, which is averaged over the ten subjects, and plotted in Figure 3(b), as normative value compared to patient 1”

Line 289: “In Figure 3(b), we found that after only four weeks it reached the baseline of the normal control group (normative value), indicating good recovery in all segments of the facial area”

Comment 7: Figure 4: it is poorly readable.

Response 7: Font size and image quality are improved for better readability.

Comment 8: Line 357: DISC's non-contact nature: what is the meaning?

Response 8:  Changed to  “DISC’s non-contact method of operation” for better clarity

Reviewer 2 Report

Comments and Suggestions for Authors

This manuscript introduces the novel application of Digital Image Speckle Correlation for tracking facial muscle activity as a non-contact, video-based diagnostic method for neurological and psychiatric disorders. The technique is compelling in its potential clinical utility and accessibility, particularly for real-time or remote settings. The study is clearly structured, applying DISC to two case studies—Bell’s palsy and treatment-resistant depression/anxiety—and comparing results with a healthy control cohort. The methodological rigor, including reaction time measurements and statistical correlations with validated psychiatric scales, lends credibility to the approach. Moreover, the use of skin pores as natural speckle markers and the application of well-established optical flow algorithms are thoughtfully implemented.

However, while the study presents interesting findings, several critical limitations should be addressed. The sample size is too small—consisting of only two patients and ten controls—to draw any generalizable conclusions. The authors claim DISC can differentiate psychiatric or neurologic pathology and even assess treatment response, yet the narrow patient sample and lack of longitudinal tracking significantly weaken that assertion. Particularly in the psychiatric case, confounding factors such as polypharmacy, long-term ketamine exposure, and individual variability in facial muscle expressivity are not sufficiently controlled or discussed. Furthermore, the paper leans heavily on visual inspection of heatmaps and summary tables without providing raw reaction time distributions or confidence intervals for critical outcomes.

The manuscript would benefit from a clearer articulation of limitations, especially concerning scalability and reproducibility. Additionally, the discussion occasionally makes speculative claims regarding DISC’s clinical impact and diagnostic capabilities without adequate evidence or comparative analysis to existing tools (e.g., sEMG or facial EMG). Minor issues include grammatical inconsistencies and occasional ambiguity in figure descriptions.

Author Response

Comment 1: The sample size is too small—consisting of only two patients and ten controls—to draw any generalizable conclusions. The authors claim DISC can differentiate psychiatric or neurologic pathology and even assess treatment response, yet the narrow patient sample and lack of longitudinal tracking significantly weaken that assertion.

Response 1: We appreciate the reviewer’s concern regarding the limited sample size. We have now substantially revised the Introduction to clarify the rationale for presenting these two case studies. These cases were not intended to establish generalizable clinical conclusions, but rather to build upon a substantial foundation of prior work validating DISC as a robust tool for assessing facial neuromuscular enervation. Previous studies from our group and others have demonstrated DISC’s ability to localize and quantify muscle activation patterns in healthy individuals, track paralysis and recovery following Botox injections, and detect subtle micro-emotional responses to calibrated visual stimuli.

In this context, the current paper represents a translational step: we selected two representative clinical cases—one neurological (Bell’s palsy) and one psychiatric (treatment-resistant depression)—to illustrate how DISC can be applied in pathological conditions. These patients were chosen deliberately because they allowed for internal controls and longitudinal comparisons. While we agree that larger, longitudinal studies are required to confirm generalizability, these two cases provide proof-of-concept demonstrations of DISC’s clinical relevance and set the stage for broader clinical validation.

The modified the introduction with a more comprehensive description as follows;

“The technique was first applied as a method for facial recognition, where we demonstrated that it was another method for determination of the muscles involved in facial expression, which are unique to the individual[10]. A much larger 18 months trial was performed on patients injected with Botox, where Bhatnagar et al demonstrated that DISC can follow the location and intensity of the paralysis as well as the subsequent recovery[11]. Verma et al demonstrated that DISC can be used to measure the intensity of motion required for a given expression, which is unique to the individual [12]. Using this information, DISC was used to determine the optimal dose of Botox required to achieve muscular paralysis. This study also showed improved recovery in patients where DISC was used to guide the injection site, relative to another group where injection was performed following standard protocols.

Most recently, Saadon et al showed that emotional response could also be detected via rapid facial muscular enervation, without any external facial expression and in this manner could be used for real-time detection and classification of micro- emotions (happy, sad, and neutral) by analyzing subtle muscular features invisible to the naked eye[13].

The application of DISC to the analysis of a specific pathology was demonstrated by Bhatnagar, where it was shown that DISC could detect impairment of the facial nerve and subsequent recovery following surgery, in patients diagnosed with vestibular schwannoma [14].

 Except for the acoustic neuroma study, the previous work selected only healthy individuals, where the experiments were conducted mostly to demonstrate the abilities of disc in neuromuscular evaluation. Hence in this paper we selected two pathologies, one involving a direct physical impairment, Bell’s Palsy, which was selected since paralysis and recovery of the affected site could be compared on the same patient.  A second patient was selected with a condition involving a mental state, where the response of a patient undergoing specific therapy could be directly compared to a much larger control cohort of healthy individuals.  We detail the findings of the technique as compared to the results obtained from a control cohort without underlying pathology to illustrate the power of this non-contact method for diagnosis, accomplished solely through video recording and programming-based analysis.

Comment 2: Particularly in the psychiatric case, confounding factors such as polypharmacy, long-term ketamine exposure, and individual variability in facial muscle expressivity are not sufficiently controlled or discussed.

Response 2: Here we only show that this particular ketamine patient exhibits delayed reaction time and increased dissociative behavior, as compared to a larger control cohort.  We also show that these observations are consistent with the self reported scores of her mental state at the time of the visits.

Comment 3: Furthermore, the paper leans heavily on visual inspection of heatmaps and summary tables without providing raw reaction time distributions or confidence intervals for critical outcomes.

Response 3: The heatmaps are shown simply to guide the eye. The reaction times were quantitatively determined from the derivative to the displacement curves, as shown in figure 4. The statement “visual reaction time” was calculated mathematically from the temporal analysis of the curves.

Comment 4: The manuscript would benefit from a clearer articulation of limitations, especially concerning scalability and reproducibility.

Response 4: The general reproducibility of the DISC technique was discussed extensively in the prior publications involving Botox injections, where patients were monitored over extended time periods of time, and in the publication involving detection of micro-emotions which were elicited in several ways following exposure to calibrated images. In our revised introduction we have included those references and provided a better description of the technique placing the paper in context of previous work, where those results obtained on healthy patients were then used to illustrate the potential of DISC in the detection of a pathology. We also emphasize that this paper only involves detection capability, and does not deal with any aspect of treatment or its evaluation. This is clearly the major limitation of this paper—since with only two patients we cannot make any statement regarding protocols. We do allude to the fact that DISC can be used to collect large amounts of data on impairment cause by different conditions, which can then be integrated with AI analysis for evaluation of different outcomes or treatments.

Comment 5: Additionally, the discussion occasionally makes speculative claims regarding DISC’s clinical impact and diagnostic capabilities without adequate evidence or comparative analysis to existing tools (e.g., sEMG or facial EMG). Minor issues include grammatical inconsistencies and occasional ambiguity in figure descriptions.

Response 5: We did not directly compare DISC to EMG since the technique is invasive and not indicated for cosmetic Borox treatments. Botox injection though was described by Bhatnagar et al is specific to certain muscles, and in the prior publication it was discussed how easily this muscle, its function, and degree of paralysis,  can imaged with DISC. It was also demonstrated the ability of DISC to map the muscular functions across the facial area, and image the redistribution of muscular activity in achieving a given facial expression, once the targeted muscle has been paralyzed with the Botox injection. Therefore, DISC provided information which would have been difficult and painful to obtain with sEMG.

Similarly, sEMG could probably detect micro-emotions, but again its not indicated for this application. In the case of Bell’s Palsy, it can be used, but its inability to map the specific location of the affected muscles does not provide useful additional information from the visible examination. In this case DISC may be more effective, since it maps the interplay of all facial muscles and can image the recovery process.

Round 2

Reviewer 1 Report

Comments and Suggestions for Authors

The Authors cite three papers in the revised version:

  • reference 11: they enrolled 6 patients;
  • ref. 12: they enrolled 10 patients;
  • ref. 13: they enrolled 10 patients.

This is to corroborate my suggestion to expand the patient cohort. I think it is unavoidable. 

Author Response

We appreciate the reviewer’s concern and agree that expanding the patient cohort would provide additional statistical power for population-level conclusions. Unfortunately, we are unable to address the reviewer’s comments in reference to our previously published papers. The variance in those papers was very tight, and the results were published in respected peer reviewed journals.  In this publication, we are applying the technique to two well defined pathologies, where there are well defined internal controls for the patients. This emphasizes the “case study” emphasis of this paper.

Our objective is not to present a statistical estimate of diagnostic accuracy across a population, but rather to demonstrate how the technique applies to individual cases. demonstrating how specific quantitative metrics—each showing significant deviation from the control cohort—can be extracted. The control cohort's statistical parameters, including the mean and standard deviation, have already been established and validated in prior studies to represent the general population distribution.

In the case of Bell’s palsy, we leveraged the internal control provided by the unaffected side of the patient’s face, tracking the recovery of asymmetry over time. In the psychiatric case, we measured significant differences in reaction time (p < 0.05) relative to control values and correlated the DISC results with standardized self-assessment scales, achieving >85% correlation in four out of six measures.

We clearly state in the revised manuscript that this is not a population study and that the findings cannot be generalized to all patients. To make this distinction explicit to readers, we added the following sentence to the beginning of the discussion section:

“It is worth noting that this paper describes two case studies, where the applicability of the technique is described in each case, based on the physical principals of the methodology. This is not a population study, and hence we cannot predict general outcomes."

Reviewer 2 Report

Comments and Suggestions for Authors

The authors addressed the comments.

Author Response

We sincerely thank you for your thoughtful and constructive comments on our manuscript. Your insights have been invaluable in helping us improve the clarity, scientific rigor, and overall quality of our work.

We have carefully considered each of your suggestions and revised the manuscript accordingly. Where appropriate, we have added clarifications, updated the text to address methodological concerns, and highlighted the limitations and scope of our findings as you recommended. We believe these revisions have strengthened the manuscript significantly.

We are grateful for your time and expertise!